# Effectiveness of Systemic Corticosteroids in Managing Cancer-Related Neuropathic Pain: A Multicenter Prospective Observational Study

**DOI:** 10.3390/cancers17101630

**Published:** 2025-05-12

**Authors:** Keita Tagami, Takaomi Kessoku, Hideaki Hasuo, Hiroto Ishiki, Takashi Yamaguchi, Masanori Mori, Yusuke Hiratsuka, Kazuhiro Kosugi, Yuka Okuda, Takeya Yamaguchi, Shingo Miyamoto, Kiyofumi Oya, Kaoru Nishijima, Yoshika Koinuma, Naoto Morikawa, Shunsuke Oyamada, Keisuke Ariyoshi, Masaki Higuchi, Hironori Mawatari, Kosuke Tanaka, Mariko Shimazu, Daisuke Kiuchi, Mamiko Sato, Michihiro Iwaki, Rintaro Koike, Kazuki Sato, Akira Inoue

**Affiliations:** 1Department of Home-Based Palliative Care, Yushoukai Home Medical Clinic Nerima, 1-22-11, Hazawa, Nerima-ku, Tokyo 176-0003, Japan; 2Department of Palliative Medicine, Tohoku University Graduate School of Medicine, Seiryo-machi, Aoba-ku, Sendai 980-8575, Miyagi, Japan; hiratsuka.med.t@gmail.com (Y.H.); 11016mmk@gmail.com (M.S.); rintaro.koike.e2@tohoku.ac.jp (R.K.); akira.inoue.b2@tohoku.ac.jp (A.I.); 3Department of Palliative Medicine and Gastroenterology, International University of Health and Welfare Narita Hospital, 852 Hatagateda, Narita 286-8250, Chiba, Japan; takaomi0027@gmail.com (T.K.); kosuke.tsssik@gmail.com (K.T.); 4Department of Gastroenterology and Hepatology, Yokohama City University School of Medicine, 1-4-1 Okina-cho, Naka-ku, Yokohama 231-0028, Kanagawa, Japan; michihirokeidai@yahoo.co.jp; 5Department of Psychosomatic Medicine, Kansai Medical University, 2-5-1 Shinmachi, Hirakata 573-1010, Osaka, Japan; h.hasuo7@gmail.com (H.H.); shimazum@hirakata.kmu.ac.jp (M.S.); 6Department of Palliative Medicine, National Cancer Center Hospital, 5-1-1, Tsukiji, Chuoku, Tokyo 104-0045, Japan; hishiki@ncc.go.jp (H.I.); dkiuchi@hosp.ncgm.go.jp (D.K.); 7Department of Palliative Care, Konan Medical Center, 1-5-16 Kamokogahara, Higashinada-ku, Kobe 658-0064, Hyogo, Japan; ikagoro@pop06.odn.ne.jp; 8Department of Palliative Medicine, Kobe University Graduate School of Medicine, 7-5-1 Kusunoki-cho, Chuo-ku, Kobe 650-0017, Hyogo, Japan; knishiji@med.kobe-u.ac.jp; 9Division of Palliative and Supportive Care, Seirei Mikatahara General Hospital, 3453 Mikatahara-cho, Chuo-ku, Hamamatsu 433-8558, Shizuoka, Japan; glacemori@hotmail.com; 10Department of Palliative Medicine, Takeda General Hospital, 3-27 Yamagamachi, Aizuwakamatsu 965-8585, Fukushima, Japan; 11Department of Palliative Medicine, National Cancer Center Hospital East, 6-5-1, Kashiwanoha, Kashiwa 277-8577, Chiba, Japan; kanabunapapa@gmail.com; 12Department of Palliative and Supportive Care, University of Tsukuba Hospital, 2-1-1, Amakubo, Tsukuba 305-8576, Ibaraki, Japan; 13Department of Anesthesiology, School of Medicine, Wakayama Medical University, 811-1 Kimiidera, Wakayama-shi 641-8509, Wakayama, Japan; yuuka-o@wakayama-med.ac.jp; 14Department of Palliative Care, JCHO Kyushu Hospital, 1-8-1 Kishinoura, Yahatanishi-ku, Kitakyushu 806-8501, Fukuoka, Japan; m04091ty@jichi.ac.jp; 15Department of Medical Oncology, Japanese Red Cross Medical Center, 4-1-22 Hiroo, Shibuya-ku, Tokyo 150-8935, Japan; aaa17580@pop06.odn.ne.jp; 16Peace Home Care Clinic Kyoto, 11-43 Kamate-cho, Yamashina-ku, Kyoto 607-8031, Kyoto, Japan; 4joekin5@hey.com; 17Department of Respiratory Medicine, Juntendo University School of Medicine, 3-1-3 Hongo, Bunkyo-ku, Tokyo 113-8431, Japan; ymatsuda@juntendo.ac.jp; 18Department of Medical Oncology, Tohoku Rosai Hospital, 4-3 Dainohara, Aoba-ku, Sendai 981-8563, Miyagi, Japan; oncology.morikawa@gmail.com; 19Department of Biostatistics, JORTC Data Center, 2-54-6-302, Nishinippori, Arakawa-ku, Tokyo 116-0013, Japan; shunsuke.oyamada@jortc.jp; 20Department of Data Management, JORTC Data Center, 2-54-6-302, Nishinippori, Arakawa-ku, Tokyo 116-0013, Japan; keisuke.ariyoshi@jortc.jp; 21Department of Palliative Care, Eiju General Hospital, 2-23-16, Higashi-Ueno, Taito-ku, Tokyo 110-8645, Japan; higuchi0515koto.a0t.skytree634@gmail.com; 22Department of Palliative and Supportive Care, Yokohama Minami Kyosai Hospital, 1-11-10, Mutsuurahigashi, Kanazawa-ku, Yokohama 236-0037, Kanagawa, Japan; hiromawa@gmail.com; 23Department of Palliative Care, Center Hospital of the National Center for Global Health and Medicine, 1-21-1, Toyama, Shinjuku-ku, Tokyo 162-8655, Japan; 24Nursing for Advanced Practice, Department of Integrated Health Sciences, Nagoya University Graduate School of Medicine, 1-1-20 Daiko-Minami, Higashi-ku, Nagoya 461-8673, Aichi, Japan; sato.kazuki.s6@f.mail.nagoya-u.ac.jp

**Keywords:** cancer pain, neuropathic pain, corticosteroids, palliative care, cancer, spinal cord, brain, inflammation, neuropathy, primary sensory neurons

## Abstract

Cancer-related neuropathic pain (CR-NP) frequently causes severe pain that impairs patients’ daily function and quality of life, remaining a critical unmet need in oncology. This multicenter prospective observational study aimed to clarify the effectiveness of systemic corticosteroids for relieving CR-NP in 107 inpatients. Our findings showed that corticosteroids provided rapid and significant relief from intense CR-NP within days, which translated into patient-reported improvements in daily functioning and sleep quality. This analgesic effect was particularly notable for severe pain originating from central nervous system involvement and was achieved without requiring increased opioid doses. Crucially, this study provides valuable foundational data regarding responsive CR-NP subtypes and potential dosing strategies where evidence was previously limited. This research would suggest that corticosteroids, demonstrated in this study to be effective and relatively safe in a one-week period, represent a valuable initial therapeutic strategy for managing CR-NP, and these foundational findings encourage further investigation into long-term effectiveness, safety, and optimized dosing strategies through future clinical trials to potentially improve sustained pain control and patient outcomes

## 1. Introduction

Pain is a major complication among patients with cancer. Managing cancer pain, including its neuropathic subset, poses a public health challenge due to its impact on quality of life (QOL), contributing to anxiety, depression, impaired sleep, and limitations in activities of daily living (ADL) [1,2,3]. A previous study reported that 54.6% of patients with advanced cancer experienced cancer pain, with 40.7% rating its intensity as ≥5 on a 0–10 numerical rating scale (NRS) [4]. A systematic review by Roberto et al. estimated that the prevalence of neuropathic pain among patients with cancer and pain is 31.2% (95% confidence interval [CI]: 27.0–35.0) [5]. Similarly, a survey of Japanese palliative care specialists reported a median prevalence of cancer-related neuropathic pain (CR-NP) of 30% (interquartile range [IQR]: 20–40) [6]. Despite its high prevalence, CR-NP remains undertreated due to its complex etiology, including tumor-induced nerve damage, mixed pain mechanisms, diagnostic challenges, and the limitations of opioids and adjuvant analgesics [1,2,7,8]. Finnerup et al. reported that the number-needed-to-treat (NNT) for neuropathic pain medications ranged from 3.6 to 7.7, highlighting their limited efficacy. These findings underscore the need for pharmacological strategies tailored to the complex pathophysiology of CR-NP [2,7].

Corticosteroids are key adjuvants in cancer pain management, valued for their anti-inflammatory and immunosuppressive properties [1,2,9,10,11,12,13,14]. International guidelines and systematic reviews recommend corticosteroids for managing cancer-related and inflammation-driven neurologic symptoms, including intracranial hypertension, brain tumor-associated edema, and tumor-induced nerve (including spinal cord) compression [1,2,11,12,13]. However, a systematic review concluded that corticosteroids provide only modest short-term pain reduction, with the quality of evidence rated as “very low” because of small sample sizes and heterogeneous methodologies [9,15]. While corticosteroids are widely used for CR-NP, evidence supporting their efficacy for specific oncologic etiologies and optimal dosing remains limited [1,10,11,12,13], necessitating further investigation.

This multicenter, prospective observational study was designed to assess the effectiveness of systemic corticosteroids for CR-NP. Furthermore, we evaluated their impact using patient-reported outcomes (PROs) and ADL measures to clarify their role in the palliation of cancer pain, including CR-NP. This study builds on prior reports on adjuvant analgesics for refractory cancer pain, including a nationwide survey of palliative care specialists, which identified corticosteroids as commonly used for specific cancer-related pathophysiological mechanisms [10].

## 2. Materials and Methods

### 2.1. Study Design

This multicenter, prospective observational study was conducted at 17 medical facilities in Japan, including specialized palliative care (SPC) units, SPC consultation teams, and oncology inpatient units, from 1 June 2020, to 31 December 2021. We enrolled inpatients aged ≥20 years with histologically confirmed cancer who initiated or escalated corticosteroids for CR-NP. To reflect real-world clinical practice, the protocol permitted the use of different systemic corticosteroids (dexamethasone, betamethasone, and prednisolone) based on physician preference and patient-specific factors. Eligible patients reported a worst CR-NP intensity of ≥4 in the past 24 h on the Japanese version of the Brief Pain Inventory–Short Form (BPI-SF), which uses a 0–10 NRS (0 = no symptoms, 10 = worst possible pain), and required additional pain management [16]. This study focused on five oncologic etiologies classified as CR-NP: (1) malignant brain tumors, (2) leptomeningeal carcinomatosis, (3) spinal cord involvement, (4) radiculopathy, and (5) peripheral nerve involvement. These conditions involve CR-NP due to tumor-related nerve invasion, compression, intracranial hypertension, or inflammation (such as peritumoral edema). A previous nationwide survey identified these as common indications for corticosteroid use in refractory cancer pain, including CR-NP [6,10].

CR-NP was diagnosed based on imaging—computed tomography (CT), magnetic resonance imaging, or positron emission tomography-CT—and physical assessments by attending physicians, incorporating pertinent physical findings indicative of neuropathic pain, followed by the application of etiology-specific diagnostic criteria [8]. For leptomeningeal carcinomatosis, cerebrospinal fluid analysis was accepted, instead of imaging. All evaluations were conducted within 30 days before enrollment. For headache-related central neuropathic pain, diagnosis required fulfillment of both criteria from the *International Classification of Headache Disorders, 3rd Edition* [17]: (1) headache developed in relation to tumor onset or led to tumor discovery, and (2) headache worsened with tumor progression. Neuropathic pain associated with spinal cord or peripheral nerve involvement, including radiculopathy, was assessed using the Japanese version of the SLANSS. A SLANSS score of ≥10, confirmed by attending physicians, established neuropathic pain [18].

Patients deemed unsuitable by their physicians were excluded. Exclusion criteria included pregnancy, breastfeeding, or possibly being pregnant; uncontrolled diabetes requiring additional insulin despite standard hypoglycemic therapy; active infection-related fever at enrollment; undergoing treatment for active peptic ulcers; and suspected or confirmed hematologic cancers, such as leukemia or malignant lymphoma. Patients were also excluded if they had a history of severe corticosteroid reactions, received corticosteroid doses exceeding the scheduled regular dose the day before enrolment, underwent surgery for pain-causing tumors or lesions within the past week, or initiated or were scheduled to initiate new anticancer therapies involving molecularly targeted drugs or immune checkpoint inhibitors within 2 weeks before or 1 week after enrollment. Patients starting these therapies after study enrollment were also excluded from the analysis.

Patients were followed for 7 days. Data were collected at baseline (T0: within 24 h after enrollment and before initiating or escalating corticosteroid therapy), at 72 h (T1), and at 168 h (T2) after T0 by attending physicians. To accommodate clinical logistics, T1 and T2 data were collected within 24 h before or after the scheduled time points. T1 was chosen based on a survey showing a median response time of 72 h for adjuvant analgesics, including corticosteroids, for refractory cancer pain [10]. T2 was selected based on previous studies indicating that corticosteroid effectiveness for cancer pain is typically evaluated over a few days to approximately 1 week [10,19,20]. Assessments for patients who discontinued treatment before T1 or between T1 and T2 were recorded as T1 or T2 assessments, respectively, ensuring that the evaluations aligned with the most relevant treatment phase.

This study was conducted in accordance with the Declaration of Helsinki and the Japanese Ministry of Health, Labor, and Welfare guidelines for medical and health research involving human participants. As a noninterventional observational study, the requirement for written informed consent was waived. This study was approved by the independent ethics committee of Tohoku University School of Medicine (Approval No.: 2019-1-856) and the institutional review board of each participating institution.

### 2.2. Longitudinal Assessment of CR-NP

At baseline (T0) and follow-up (T1 and T2), participants rated their average and worst CR-NP pain intensities over the past 24 h using the NRS [16]. Pain interference with daily activities was measured using the BPI-SF, and sleep interference was assessed using the DSIS, ranging from 0 (no interference) to 10 (unable to sleep due to pain) [21,22,23,24]. The PPG was set at T0 by asking, “At what level would you feel comfortable with pain?” [25,26,27,28]. The PGIC was assessed at T1 and T2, with ratings ranging from 0 (very much improved) to 7 (very much worse) [24,29].

### 2.3. General Baseline Assessment, Analgesic Medications, and Adverse Events

At T0, we recorded the pathophysiological mechanism of CR-NP and the location of the most severe pain. The use of analgesics, analgesic adjuvants, and osmotic diuretics for pain palliation was also noted. The type, scheduled daily dose, and administration route of corticosteroids and opioids were recorded at T0, T1, and T2. Regular daily doses of corticosteroids were converted to a DEDD, and opioids to an oral MEDD. The conversion ratio for DEDD used in this study, based on anti-inflammatory potency [30], was as follows: hydrocortisone 100 mg = prednisolone 25 mg = methylprednisolone 25 mg = betamethasone 4 mg = dexamethasone 4 mg. The widely practiced MEDD conversion ratios in Japan [31,32] were as follows: oral morphine 60 mg = intravenous/subcutaneous morphine 30 mg = epidural morphine 3 mg = intrathecal morphine 0.3 mg = oral oxycodone 40 mg = intravenous/subcutaneous oxycodone 30 mg = fentanyl patch 25 μg/h = intravenous/subcutaneous fentanyl 0.6 mg = oral tapentadol 200 mg = oral hydromorphone 12 mg = intravenous/subcutaneous hydromorphone 2.4 mg = oral methadone 6 mg = oral/intravenous tramadol 300 mg = oral codeine 360 mg. During the observation period, discontinuation or dose reduction of corticosteroids, along with the reasons, were recorded. Changes in other regular analgesics, analgesic adjuvants, or osmotic diuretics, including dose adjustments (none, increase, new start, decrease, or discontinuation), were also recorded at T1 and T2.

Baseline demographic characteristics, such as age, biological sex, primary cancer site, Charlson Comorbidity Index, and ongoing anticancer treatment, such as chemotherapy, radiation therapy, and chemo-radiotherapy, were collected at T0 [33]. Any new anticancer therapies initiated after T0 were recorded at T1 and T2. Instrumental ADLs were assessed using the AKPS at T0, T1, and T2 [34]. The AKPS assigns scores from 0 to 100 (in 10-point increments) based on the patient’s ability to perform daily tasks and was developed for palliative care populations [34].

The severity of corticosteroid-related adverse events, such as hyperglycemia, insomnia, delirium, gastrointestinal bleeding, and perforation, was recorded and rated from Grade 1 to 5 based on the Japanese version of the CTCAE version 5.0. Causality was assessed as possible, probable, or definite [35]. If an adverse event occurred more than once, the highest severity was reported.

### 2.4. Data Analyses

Descriptive statistics, including counts, proportions (percentages), means with SDs, SEs, 95% CIs, and medians with IQRs or ranges, were used to summarize the data.

The primary endpoint was the change in the worst and average CR-NP intensity scores on the NRS from T0 to T1 and T2. Differences in mean values were assessed using paired *t*-tests. The analysis was also stratified by specific pathological CR-NP etiologies: (1) malignant brain tumors, (2) leptomeningeal carcinomatosis, (3) spinal cord involvement, (4) radiculopathy, and (5) peripheral nerve involvement. Linear regression analysis was performed to assess the association between changes in MEDD and the worst CR-NP intensity score on NRS from T0 to T1 and T2.

Secondary outcomes included changes in pain interference with general activities and DSIS scores, compared from T0 to T1 and T2 using paired *t*-tests. Pearson correlation coefficients were calculated to assess the relationship between changes in AKPS, and both the mean and worst CR-NP intensity scores on NRS from T0 to T1 and T2. Fisher’s z-transformation was used to calculate the CI. Descriptive statistics were used to evaluate the proportion of patients who achieved a 33%, 50%, or 100% reduction in the worst and average CR-NP intensity scores on the NRS, as well as their PPG at T1 and T2. Achievement of PPG was defined as an average CR-NP intensity score on the NRS ≤ the PPG score, as determined at T0 [25,26].

Only existing data were analyzed, without imputation. A significance level of 0.05 (two-sided) was used. All analyses were performed with SAS Version 9.4 (SAS Institute, Cary, NC, USA).

## 3. Results

Of the 108 screened patients with cancer, 107 were enrolled and analyzed. At T1, 5 patients, and at T2, 10 patients had missing data or discontinued corticosteroid therapy (Figure 1). Among the 107 patients at T0, CR-NP was categorized as follows: 27 patients with malignant brain tumors, 7 with leptomeningeal carcinomatosis, 22 with spinal cord tumor involvement, 30 with radiculopathy, and 21 with peripheral nerve involvement (Figure 1).

### 3.1. Demographic and Baseline Clinical Characteristics

Table 1 presents the demographic data of 107 participants. The mean age was 62.6 years (standard deviation [SD], 13.2), and 57 (53.3%) were female. Lung cancer was the most common primary site (25.2%). Comorbidities were present in 29 (27.1%) patients, including 10 with diabetes mellitus. Ongoing anticancer treatment was reported in 34 (31.8%) patients, with no new treatments initiated during the study period.

Table 2 summarizes the baseline cancer pain and clinical characteristics. The most common pain site was the head and neck (37 patients, 34.6%). The mean Australian Karnofsky Performance Scale (AKPS) at T0 was 47.3 (SD, 18.4), and the median personalized pain goal (PPG) score was 3 (IQR, 2–4). The median initial corticosteroid dose was 6.6 mg (IQR, 4–8), with 25 patients requiring an increased dose for CR-NP management. Dexamethasone was the most used corticosteroid (78 patients, 72.9%), followed by betamethasone (21 patients, 19.6%), administered primarily intravenously or orally. Regular opioids were prescribed to 72 (67.3%) patients, with a median morphine equivalent daily dose (MEDD) of 47.5 mg (IQR, 30–120). Among 84 patients receiving non-opioid analgesics, adjuvant analgesics (excluding corticosteroids), or diuretics, 47 (43.9%) used non-steroidal anti-inflammatory drugs (NSAIDs), 28 (26.2%) received gabapentinoids, and 16 (15.0%) were on osmotic diuretics.

### 3.2. Changes in CR-NP Intensity and Pain Interference with Activities and Sleep

Table 3 presents the changes in the worst and average CR-NP intensity scores, pain interference with general activities, and daily sleep interference score (DSIS) from T0 to T1 and T2. Both the worst and average CR-NP intensity decreased substantially over time. The mean worst CR-NP intensity decreased from 8.2 (SD, 1.9) at T0 to 5.2 (SD, 2.9) at T1 and 4.4 (SD, 3.0) at T2, corresponding to differences of −3.0 (95% CI −3.6 to −2.4) and −3.8 (95% CI −4.5 to −3.3) (*p* < 0.01 for both), respectively. Similarly, the mean average CR-NP intensity decreased from 5.8 (SD, 2.2) at T0 to 3.5 (SD, 2.4) at T1 and 3.0 (SD, 2.5) at T2, with differences of −2.3 (95% CI −2.8 to −1.7) and −2.8 (95% CI −3.3 to −2.2) (*p* < 0.01), respectively. Pain interference with general activities and DSIS also improved significantly. Pain interference with general activities declined by −2.7 (95% CI −3.3 to −2.1) from T0 to T1 and −3.4 (95% CI −4.0 to −2.8) from T0 to T2. The DSIS decreased by −2.7 (95% CI −3.2 to −1.9) from T0 to T1 and −3.4 (95% CI −4.0 to −2.6) from T0 to T2.

### 3.3. Proportions of Pain Reduction (33%, 50%, and 100%) and Achievement of PPG and Patient Global Impression of Change (PGIC)

Appendix A Table A1 presents the proportion of patients achieving 33%, 50%, and 100% pain reduction and their PPG scores. At T1, 54.9% of patients experienced a 33% reduction in the worst pain, increasing to 66.0% at T2, while 56.3% experienced a 33% reduction in average pain, rising to 71.4% at T2. A 50% reduction in worst pain was observed in 39.2% at T1 and 55.7% at T2, while 46.6% achieved a 50% reduction in average pain at T1, increasing to 56.1% at T2. Complete pain relief (100% reduction) was reported in 11.7% at T1 and 20.4% at T2. PPG scores set at T0 were achieved by 51.0% of patients at T1 and 60.0% at T2. Appendix A Table A2 presents PGIC results for corticosteroid treatment of CR-NP. At T1, 79.6% of patients reported improvement (very much, much, or minimally improved), rising slightly to 79.8% at T2.

### 3.4. Changes in CR-NP Intensity by Pathological Mechanism

Figure 2 illustrates substantial reductions in the worst and average CR-NP intensity scores from T0 to T2 across various pathological mechanisms. For the worst pain intensity, patients with malignant brain tumors experienced a decrease from 7.9 (SD, 2.1) at T0 to 3.8 (SD, 2.5) at T1 and 2.6 (SD, 2.6) at T2, with differences of −4.1 (95% CI −5.1 to −2.9) and −5.2 (95% CI −6.2 to −4.2), respectively, both with *p* < 0.01. In leptomeningeal carcinomatosis, the worst pain intensity declined from 9.0 (SD, 1.0) at T0 to 5.0 (SD, 2.0) at T1 and 3.8 (SD, 2.6) at T2, with differences of –4.0 (95% CI −6.3 to −1.7) and −5.3 (95% CI −8.1 to −2.5), respectively, both with *p* < 0.01. Patients with spinal cord involvement showed a reduction from 8.8 (SD, 1.4) at T0 to 5.8 (SD, 3.2) at T1 and 4.9 (SD, 3.2) at T2, with differences of −3.0 (95% CI −4.4 to −1.6) and −3.9 (95% CI −5.4 to −2.4), respectively, both with *p* < 0.01. In patients with radiculopathy, the worst pain intensity decreased from 8.5 (SD, 1.9) at T0 to 5.5 (SD, 2.8) at T1 and 5.1 (SD, 2.9) at T2, with differences of −3.0 (95% CI −4.3 to −2.1) and −3.4 (95% CI −4.9 to −2.5), respectively, both with *p* < 0.01. Patients with peripheral nerve involvement exhibited a decrease from 7.4 (SD, 2.2) at T0 to 6.0 (SD, 2.7) at T1 and 5.6 (SD, 2.7) at T2, with differences of −1.4 (95% CI −2.6 to −0.1; *p* = 0.04) and −1.8 (95% CI −3.3 to −0.5; *p* = 0.01), respectively (Figure 2a).

Similarly, the average CR-NP intensity considerably decreased across all pathological mechanisms (Figure 2b). Among patients with malignant brain tumors, the average pain intensity declined from 5.7 (SD, 2.5) at T0 to 2.7 (SD, 2.5) at T1 and 1.8 (SD, 2.4) at T2, with differences of −2.8 (95% CI −4.1 to −1.6) and −3.7 (95% CI −4.8 to −2.7), respectively, both with *p* < 0.01. In leptomeningeal carcinomatosis, the average pain intensity decreased from 6.9 (SD, 2.1) at T0 to 3.8 (SD, 2.3) at T1 and 3.5 (SD, 1.9) at T2, with differences of −2.8 (95% CI −4.4 to −1.2) and −3.3 (95% CI −4.8 to −1.9), respectively, both with *p* < 0.01. Patients with spinal cord involvement showed a reduction from 5.9 (SD, 2.0) at T0 to 3.5 (SD, 3.0) at T1 and 2.8 (SD, 2.8) at T2, with differences of −2.4 (95% CI −4.0 to −0.7) and −3.0 (95% CI −4.5 to −1.6), respectively, both with *p* < 0.01. In radiculopathy, average pain intensity decreased from 5.8 (SD, 2.0) at T0 to 3.5 (SD, 1.9) at T1 and 3.3 (SD, 2.0) at T2, with differences of −2.3 (95% CI −3.2 to −1.5) and −2.6 (95% CI −3.5 to −1.7), respectively, both with *p* < 0.01. Among patients with peripheral nerve involvement, average pain intensity decreased from 5.2 (SD, 2.3) at T0 to 4.1 (SD, 2.4) at T1 and 4.2 (SD, 2.4) at T2, with differences of −1.2 (95% CI −2.3 to −0.1; *p* = 0.04) and −1.2 (95% CI −2.4 to 0; *p* = 0.06), respectively (Figure 2b).

### 3.5. Opioid Dose Adjustments and the Correlation Between Pain Intensity and Changes in Analgesics and Adjuvant Analgesics

The opioid dosages, calculated as MEDD, were 52.5 mg (IQR, 30–112.5) at T1 and 50.0 mg (IQR, 30–112.5) at T2. Linear regression analysis revealed no statistically significant correlation between changes in opioid doses and reductions in worst pain scores on the NRS. From T0 to T1, the regression coefficient was β < 0.01 (standard error [SE] = 0.01, R^2^ = −0.01, *p* = 0.71), while from T0 to T2, it was β = 0.01 (SE = 0.01, R^2^ = 0.02, *p* = 0.14). During the observation period, four patients either increased or initiated non-opioid analgesics, while two reduced or discontinued them. Twelve patients either increased or initiated analgesic adjuvants (excluding corticosteroids), while one reduced or discontinued them. For diuretics, one patient increased or initiated them, whereas three reduced or discontinued them.

### 3.6. Relationship Between CR-NP Intensity and ADLs on the AKPS

The mean AKPS score was 56.6 (SD, 18.6) at T1 and 59.6 (SD, 19.2) at T2. Decreases in the mean and worst background CR-NP intensity scores on the NRS from T0 to T1 and T0 to T2 were significantly correlated with increases in AKPS scores. From T0 to T1, the worst pain intensity had a correlation coefficient of r = −0.38 (95% CI −0.53 to −0.19, *p* < 0.01), while the average pain intensity had a correlation coefficient of r = −0.33 (95% CI −0.49 to −0.14, *p* < 0.01). From T0 to T2, the worst pain intensity exhibited r = −0.36 (95% CI −0.52 to −0.17, *p* < 0.01), and average pain intensity had r = −0.39 (95% CI −0.55 to −0.21, *p* < 0.01).

### 3.7. Changes in Corticosteroid Dosage

The corticosteroid doses, calculated as dexamethasone-equivalent daily dose (DEDD), were 6.6 mg (IQR, 4–8) at baseline, 6.0 mg (IQR, 4–8) at T1, and 4.0 mg (IQR, 3.3–8) at T2. During the observation period, corticosteroid treatment was completed or discontinued in 16 patients. Among them, five were discontinued due to follow-up termination (discharge or other reasons), four due to lack of efficacy, three due to symptom improvement, and two due to adverse effects. In addition, corticosteroid doses were reduced in 33 patients. Of these, 20 underwent scheduled dose reductions to prevent adverse effects, despite the corticosteroids being effective for CR-NP. Nine patients had dose reductions due to lack of efficacy, while three reduced their doses due to adverse effects, even though the corticosteroid remained effective. Some patients who initially reduced their dose later discontinued or completed corticosteroid treatment, leading to some overlap in classification.

### 3.8. Adverse Events Related to Corticosteroid Treatment

During the study period, 21 corticosteroid-related adverse events were reported (Appendix A Table A3). Insomnia, delirium, and hyperglycemia occurred in 10, 8, and 3 cases, respectively. No cases of gastrointestinal bleeding, perforation, or other adverse events defined by Common Terminology Criteria for Adverse Events (CTCAE) version 5.0 were observed.

## 4. Discussion

This exploratory study examined corticosteroid therapy for CR-NP based on previous studies and clinical experiences [1,10,11,12,13,15]. We evaluated corticosteroid effectiveness in managing malignant brain tumors, leptomeningeal carcinomatosis, spinal cord involvement (including disseminated lesions), radiculopathy, and peripheral nerve involvement. Using a prospective design, we assessed treatment outcomes through PROs and quantitative scales, measuring pain intensity, impact on ADLs, and patient satisfaction while adjusting for potential confounders. Comprehensive assessments, including PROs, are crucial in cancer pain management as they address not only pain reduction, but also improvements in ADLs and overall comfort. This study systematically evaluated key treatment factors, such as optimal dosage, onset of therapeutic effect, adverse events, and corticosteroid safety in CR-NP palliation. These findings contribute to an evidence-based framework for corticosteroid use in cancer pain management, including CR-NP.

A randomized controlled trial (RCT) by Ashar et al. assessed corticosteroid efficacy for generalized cancer pain, but faced enrollment challenges due to strict eligibility criteria, such as recent corticosteroid use, unstable analgesia, and complex requirements [36]. In contrast, our multicenter prospective observational study, conducted across 17 facilities, included a more diverse cohort of 107 participants. To ensure validity despite broader inclusion criteria, we applied strict diagnostic measures, including imaging confirmation and Self-Reported Leeds Assessment of Neuropathic Symptoms and Signs (SLANSS) scoring to confirm neuropathic pain.

Although corticosteroids are commonly used in cancer pain management, their optimal dosing, treatment protocols, and the specific cancer pain subtypes most responsive to corticosteroids remain unclear. Previous reviews and guidelines have recommended corticosteroids as adjunct analgesics for certain cancer pain conditions; however, the quality of evidence supporting their use in cancer pain management has been rated as low to moderate, suggesting the need for further investigation [1,15,19,37,38]. In previous studies, corticosteroids were often combined with other treatments, such as radiotherapy for bone metastases or vertebroplasty for vertebral neoplasms [15,39,40,41]. In contrast, RCTs by Mercadante et al., Paulsen et al., and Yennurajalingam et al., targeting generalized cancer pain, failed to show effectiveness of corticosteroids [19,42,43]. Paulsen et al. excluded severe neuropathic pain (NRS ≥ 8) and central nervous system (CNS)-related cases, such as pain from brain tumors or spinal cord compression [19]. The 2018 World Health Organization guidelines suggest that corticosteroids may be particularly effective for cancer pain with inflammatory mechanisms, bone pain, neuropathic pain, and visceral pain, rather than for general cancer pain [1]. In addition, the Congress of Neurological Surgeons, American Association of Neurological Surgeons, American Society of Clinical Oncology, and Society for Neuro-Oncology recommend corticosteroids (specifically dexamethasone) for temporary symptomatic relief in patients with neurological symptoms related to mass effects from brain metastases [12,13]. These findings highlight methodological inconsistencies across previous studies, such as differences in target populations, types of cancer pain, and treatments, which may explain variability in observed effectiveness. Our study focused on high-severity CR-NP, including cases with CNS involvement, a subgroup likely to benefit from corticosteroids due to their anti-inflammatory and compressive pathophysiology. Furthermore, our findings indicate that CR-NP linked to CNS or spinal cord involvement, which initially presented with higher baseline pain intensity, showed more pronounced pain relief compared with CR-NP associated with peripheral nerve involvement.

Substantial variability and heterogeneity across previous studies—such as differences in target populations, cancer pain causes, combined treatments, and other factors—has led to wide variations in corticosteroid dosing strategies for cancer pain, leaving the optimal dose unclear [1,15]. In non-inferiority RCTs, daily administration of dexamethasone at 8 mg and methylprednisolone at 32 mg alongside opioids to patients with generalized cancer pain showed no considerable pain reduction [19,42,43]. In contrast, trials targeting specific cancer pain subtypes demonstrated effective and tolerable corticosteroid dosing regimens. Yousef et al. reported that methylprednisolone at 5 mg/kg on the day before radiotherapy prevented radiation-induced pain flares and improved motor function [40]. Chow et al. showed that oral dexamethasone administration at 8 mg/day prevented radiation-induced pain flares [41]. Our multicenter study established foundational data for optimal corticosteroid dosing in CR-NP, with a median initial dose of 6.6 mg/day (DEDD, IQR 4.0–8.0 mg) and 4.0 mg/day (IQR, 3.3–8.0 mg) after 7 days, suggesting a tolerable and potentially effective balance.

Previous reviews have emphasized the need for multifaceted approaches beyond pain reduction, advocating for PROs [1,15,38]. Accordingly, our assessment not only measured pain intensity reduction, but also evaluated multiple PROs, such as patient satisfaction, achievement of PPG, and improvements in ADLs via AKPS scores. These findings confirm the effectiveness of corticosteroids for CR-NP. The anti-inflammatory mechanism of corticosteroids, which alleviates tumor-induced inflammation and subsequent edema, may also improve nerve conduction in damaged motor and sensory neurons, directly contributing to ADL improvements [44,45]. In addition, RCTs by Paulsen et al. and Yennurajalingam et al. demonstrated considerable relief from fatigue in patients with cancer receiving corticosteroid treatment [19,20,43]. These findings suggest that corticosteroids may have multifaceted effects, including improvements in fatigue, pain, and physical function, potentially influencing various outcome measures. To ensure the validity of our study on CR-NP relief by corticosteroids, we specifically assessed PROs related to pain symptom changes. Changes in other analgesics, including MEDD and anticancer treatments, did not substantially impact CR-NP improvement. These findings highlight the comprehensive benefits of corticosteroids for CR-NP, minimizing potential bias in attributing improvements.

In this study, corticosteroid-related adverse events were few and mainly limited to insomnia, delirium, and hyperglycemia. Most events were grade 2 or lower according to CTCAE version 5.0 and were manageable, although one case involved grade ≥ 3 delirium and hyperglycemia. These findings align with previous reviews and suggest that short-term (up to 7 days) corticosteroid treatment at moderate doses (equivalent to methylprednisolone 32 mg or dexamethasone 8 mg daily) is generally well-tolerated [15,37]. The need to balance symptomatic relief with potential adverse effects, such as immunosuppression, myopathy, and hyperglycemia—especially during prolonged use—is widely acknowledged [1,2,9,11,12,13]. A previous study found an increased risk of gastrointestinal bleeding and sepsis within 5–30 days of initiating prednisone 10 mg daily for < 14 days, compared with non-steroid users, underscoring the importance of close monitoring [46]. Paulsen et al. associated high corticosteroid doses (equivalent to methylprednisolone 125 mg daily) and long-term use (over 8 weeks) with elevated adverse event rates and mortality [37]. An et al. reported that while mild adverse events in their RCTs mainly occurred within the first 2 weeks, moderate to severe events appeared later [47]. Given the 7-day observation period and limited sample size in this study, more adverse events may have occurred with longer-term corticosteroid treatment.

This study has some limitations worth noting. First, as a prospective observational study rather than an RCT, we could not rigorously control for confounding factors or compare corticosteroid effectiveness with other analgesic measures or a placebo. Furthermore, since we aimed to gather foundational data for future research and treatment development, we did not set a predetermined sample size, but instead collected cases during the registration period. This approach may have resulted in a small sample size, which is insufficient for fully exploring subgroup differences. Second, although strict eligibility criteria and multidimensional PRO assessments minimized confounding influences, pain relief may still have been influenced by unadjusted factors. Third, for patients who did not achieve pain relief, we could not rule out insufficient corticosteroid doses or baseline analgesics, such as opioids, which may have contributed to suboptimal outcomes. Furthermore, although the overall median doses of corticosteroids tended to decrease or stabilize between T1 and T2, we cannot exclude the possibility that a subgroup of patients requiring dose escalation might have experienced more favorable pain relief. Fourth, due to the multicenter prospective observational study design, variability in cancer pain management practices among researchers across study centers may have affected the scheduled corticosteroid doses, administration methods, and adjustments to other analgesics. Additionally, we did not assess potential center-specific effects across the 17 participating centers on treatment patterns and outcomes, acknowledging this as a further limitation inherent to multicenter observational design. Fifth, opioid analyses were limited to scheduled MEDD calculations, and did not account for rescue analgesic use, which could have confounded corticosteroids-related pain relief assessments. Finally, although we demonstrated corticosteroid effectiveness and safety for CR-NP over 7 days, long-term efficacy and outcomes following potential loss of effect or subsequent dose adjustments were not examined. Lastly, the reliance on patient recall over a 24 h period to report pain relief and PROs may have introduced recall bias, affecting the accuracy of the data.

## 5. Conclusions

Despite some limitations, our study demonstrates that corticosteroids offer rapid analgesic and functional benefits for CR-NP, with considerable effectiveness observed in specific CR-NP subtypes. This contrasts with the limited effectiveness reported for generalized cancer pain in previous studies [15,19,37,42,43]. These findings suggest the potential for personalized treatment approaches tailored to specific CR-NP pathological mechanisms, which could enhance individualized cancer pain management. Furthermore, corticosteroids in this study not only reduced pain intensity, but also improved ADLs. These findings contribute to a refined understanding of cancer pain management with corticosteroids, providing foundational data for future clinical research through our exploratory approach. Further studies, particularly RCTs incorporating control or placebo groups, are necessary to validate these findings, define optimal dosing, assess improvements in QOL, and ensure both effectiveness and safety. Future efforts should also focus on confirming the long-term safety and sustained therapeutic effectiveness of tailored corticosteroid treatment for CR-NP in each pathological subtype.

## Figures and Tables

**Figure 1 cancers-17-01630-f001:**
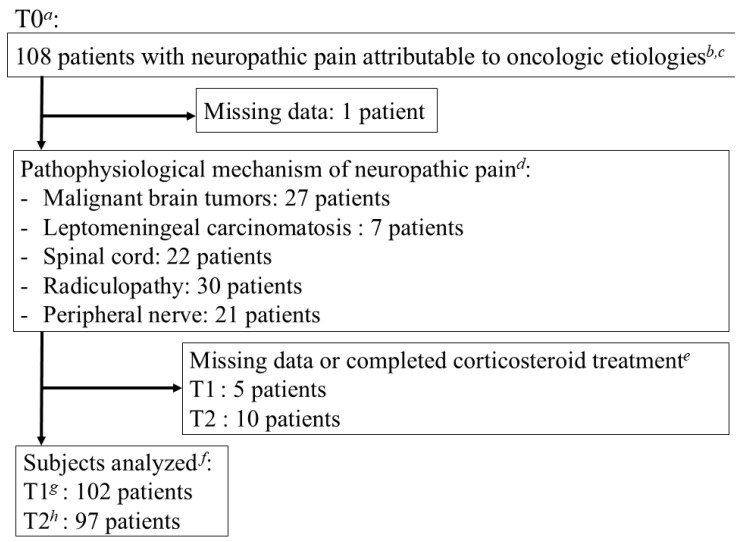
Flow chart of study population. *^a^* Within 24 h after enrollment and before starting or increasing corticosteroid treatment. *^b^* The 108 patients with cancer screened who started or increased the dose of corticosteroids for pain relief caused by central or peripheral neuropathic pain associated with oncologic etiologies. *^c^* Diagnosed at enrollment by a physician. *^d^* Including primary and metastatic tumors, as well as tumor-induced nerve invasion and compression. *^e^* During the observation period, 16 cases of corticosteroid treatment discontinuation were noted for the following reasons, excluding cases missing data: 5 cases due to completion of follow-up (discharge to home, transfer, or death), 4 cases due to lack of effectiveness for pain, 3 cases due to complete symptom relief, 2 cases due to adverse events, and 2 cases due to other reasons. *^f^* The primary endpoint was the change in the worst and average background pain intensity scores from T0 to T1 and T2. *^g^* 72 h post-enrollment and assessed within 24 h. *^h^* 168 h post-enrollment and assessed within 24 h.

**Figure 2 cancers-17-01630-f002:**
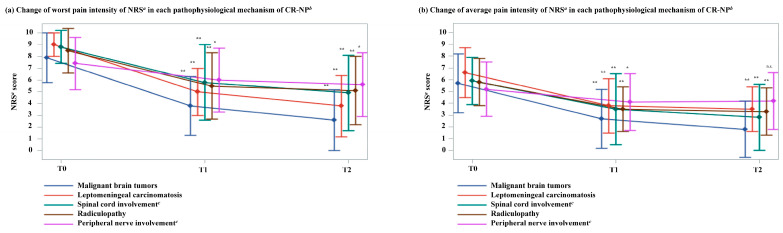
Changes in pain intensity by oncologic etiologies of neuropathic pain. (**a**) Change in worst pain intensity on the NRS *^a^* by the pathophysiological mechanism of CR-NP *^b^*; (**b**) Change in average pain intensity on the NRS *^a^* by the pathophysiological mechanism of CR-NP *^b^.* All data are presented as the mean ± standard deviation (paired *t*-tests, *, 0.01 ≤ *p* < 0.05; **, *p* < 0.01; n.s. (not significant), *p* > 0.05.). *^a^* NRS, numerical rating scale ranging from 0 to 10. *^b^* CR-NP, cancer-related neuropathic pain. *^c^* Including primary and metastatic tumors, infiltration, and compression. Vertical axis: Pain intensity score on the NRS. Horizontal axis: Time from enrollment and initiation or dose increase of corticosteroid to T1 (72 h post-enrollment) and T2 (168 h post-enrollment).

**Table 1 cancers-17-01630-t001:** Participant characteristics.

Variables (*n* = 107)	*n* (%)
Age (years)	
Mean (standard deviation)	62.6 (13.2)
Sex	
Male	50 (46.7)
Female	57 (53.3)
Primary cancer sites	
Lung	27 (25.2)
Gastrointestinal *^a^*	18 (16.8)
Breast	13 (12.1)
Gynecological *^b^*	13 (12.1)
Head and neck	10 (9.3)
Urinary *^c^*	7 (6.5)
Pancreas	4 (3.7)
Brain	4 (3.7)
Others	11 (10.3)
Charlson comorbidity index *^d^*	
None	78 (72.9)
Diabetes mellitus	10 (9.3)
Chronic obstructive pulmonary disease	4 (3.7)
Liver disease	4 (3.7)
Hemiplegia	4 (3.7)
Others	11
Anticancer treatment	
Ongoing *^d^*^,*e*^	34 (31.8)
Only observation or forgoing anticancer treatment	73 (68.2)
Starting of new anticancer treatment *^e^* during the observation period	0 (0)

*^a^* Esophagus, stomach, colon, and rectum; *^b^* Ovary and uterus; *^c^* Kidney, ureter, bladder, and prostate; *^d^* Total number of choices; *^e^* Radiation therapy, chemotherapy, and chemoradiotherapy.

**Table 2 cancers-17-01630-t002:** Baseline cancer pain and clinical characteristics.

Variables (*n* = 107)	Cases (%)
Most painful site	
Head and neck	37 (34.6)
Arm and shoulder	21 (19.6)
Leg	15 (14.0)
Back	13 (12.2)
Hip and genitals	9 (8.4)
Others	12 (11.2)
SLANSS *^a^*^,*b*^ score	
Median (IQR)	15 (12–18)
AKPS *^c^*	
Mean (SD)	47.3 (18.4)
PPG	
Median (IQR) *^i^*	3 (2–4)
Initial corticosteroid administration Dosage *^d^*^,*e*^	
Median (IQR)	6.6 (4–8)
Type of corticosteroids	
Dexamethasone	78 (72.9)
Betamethasone	21 (19.6)
Prednisolone	8 (7.5)
Route of corticosteroids	
Intravenous	53 (49.5)
Oral	50 (46.7)
Subcutaneous	4 (3.7)
Regular analgesic, analgesic adjuvants, and diuretics medication at T0 *^f^*^,*g*,*h*^	
Opioids	72 (67.3)
NSAIDs	47 (43.9)
Acetaminophen	46 (43.0)
Gabapentinoids *^k^^,j^*	28 (26.2)
Osmotic diuretics	16 (15.0)
Antidepressants	5 (4.7)
Ketamine	2 (1.9)
Others	2 (1.9)
Dosage of regular opioids at T0 *^j^*^,*k*^	
Median (IQR)	47.5 (30–120)

Abbreviations: AKPS, Australia-modified Karnofsky Performance Status; IQR, interquartile range; NSAIDs, non-steroidal anti-inflammatory drugs; PPG, personalized pain goal, SD, standard deviation; SLANSS, Self-Reported Leeds Assessment of Neuropathic Symptoms and Signs. *^a^* A total score of 10 or higher, as determined through direct evaluation by physicians, indicates peripheral neuropathic pain. *^b^* A total of 73 patients diagnosed with neuropathic pain due to spinal cord and peripheral nerve involvement, including radiculopathy, were assessed using SLANSS. *^c^* The number of missing data points was 3. *^d^* Twenty-five patients were already using corticosteroids at the beginning of the observation period. *^e^* The corticosteroids included both new regimens and those previously used and increased. *^f^* Analgesic adjuvants other than corticosteroids. *^g^* Diuretics used for palliation of pain due to malignant tumor-induced edema. *^h^* Total number of choices. *^i^* Pregabalin, gabapentin, and mirogabalin. *^j^* Morphine equivalent daily dose (mg, oral). *^k^* Seventy-two patients were treated with regular opioids.

**Table 3 cancers-17-01630-t003:** Change in brief pain inventory (BPI) short form and daily sleep interference score (DSIS).

BPI and DSIS Items ^a^	From T0 ^b^ to T1 ^c^			
	T0, Mean (SD)	T1, Mean (SD)	Difference in Means (95% CI)	*p*
Worst pain intensity in the last 24 h	8.2 (1.9)	5.2 (2.9)	−3.0 (−3.6 to −2.4)	<0.01
Average pain intensity in the last 24 h	5.8 (2.2)	3.5 (2.4)	−2.3(−2.8 to −1.7)	<0.01
Pain interference general activities	6.9 (2.5)	4.1 (2.7)	−2.7 (−3.3 to −2.1)	<0.01
Pain interference sleep	5.8 (3.1)	3.1 (2.7)	−2.7 (−3.2 to −1.9)	<0.01
**BPI and DSIS Items ^a^**	**From T0 to T2 ^d^**			
	**T0, Mean (SD)**	**T2, Mean (SD)**	**Difference in Means (95% CI)**	** *p* **
Worst pain intensity in the last 24 h	8.2 (1.9)	4.4 (3.0)	−3.8 (−4.5 to −3.3)	<0.01
Average pain intensity in the last 24 h	5.8 (2.2)	3.0 (2.5)	−2.8(−3.3 to −2.2)	<0.01
Pain interference general activities	6.9 (2.5)	3.4 (2.8)	−3.4 (−4.0 to −2.8)	<0.01
Pain interference sleep	5.8 (3.1)	2.4 (2.6)	−3.4 (−4.0 to −2.6)	<0.01

Abbreviations: CI, confidence interval; SD, standard deviation. ^a^ Ranging from 0 (no symptoms) to 10 (worst possible pain) on the numerical rating scale. ^b^ Within 24 h after enrollment and before starting or increasing corticosteroid treatment. ^c^ 72 h post-enrollment and assessed within 24 h. ^d^ 168 h post-enrollment and assessed within 24 h.

## Data Availability

The data supporting the findings of this study are available from the corresponding author, Keita Tagami, upon reasonable request. All authors consent to the journal reviewing the data if necessary.

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
