# Peer review of "Effectiveness of Systemic Corticosteroids in Managing Cancer-Related Neuropathic Pain: A Multicenter Prospective Observational Study"

_cancers, 2025, doi:10.3390/cancers17101630_

Round 1
Reviewer 1 Report
Comments and Suggestions for Authors
This clinically important and interesting prospective but observative study about the effects of different doses of various corticosteroids on neuropathic cancer pain. A total of 107 patients were followed for 7ds after they were put on corticosteroids. 72 patients were treated with dexamethason, 21 with betamethason and 8 prednisolone. Th authors observed a significant reduction in pain. The side-effects were mild. No correlations was observed between opioid dose and pain relief.
The paper is well-written and the results are important for all doctors.
The study has some serious flaws, which the authors have well recognized in the discussion-section. The problems are
1) no controll-group was collected and no placebo was used, which make strictly the results only correlative. Please, explore this.
2) how was the group-size defined and the number of individuals was only 107 patients?
3) did any of the patients have surgery or radiation therapy?
4) why different corticosteroids were allowed?
5) how many centers participated in the study?
Author Response
We are grateful for the reviewer’s comments on our manuscript and believe that incorporating the advice therein into the revised version has improved the manuscript.
Our alterations to the manuscript are listed in the following point-by-point responses to the reviewers’ comments. In addition, we have rechecked our manuscript for conformance with the formatting guidelines (including References) and corrected some language issues (The reviewer’s comments are in blue font, and the revised text in the manuscript is in red font).
Point 1: no controll-group was collected and no placebo was used, which make strictly the results only correlative. Please, explore this.
Response: Thank you for pointing out this important limitation inherent in our study design. We acknowledge that the absence of a control or placebo group prevents definitive conclusions about causality and limits our findings to correlations, as stated in the Limitations section of our Discussion.
Given the limited existing evidence specifically supporting corticosteroid efficacy across the diverse etiologies of cancer-related neuropathic pain (CR-NP) investigated in this study, a crucial first step was to gather foundational real-world data. Furthermore, proceeding directly to a randomized controlled trial (RCT) with a placebo arm, without such preliminary foundational data, required careful consideration of methodological and ethical challenges, particularly regarding patient safety and the severity of pain in this population. Therefore, this prospective observational study was methodologically designed as an exploratory investigation to assess the effectiveness and safety profile of corticosteroids as used in clinical practice.
We believe these findings provide a valuable foundation for designing more rigorous RCTs to definitively establish efficacy and guide optimal use of corticosteroids for CR-NP.
Point 2: how was the group-size defined and the number of individuals was only 107 patients?
Response: Thank you for this question regarding the sample size. As mentioned in the Limitations section of our Discussion, this study was exploratory in nature, aiming to gather foundational data for future research. Therefore, we did not perform a formal sample size calculation a priori. Instead, we consecutively enrolled eligible patients during the defined study period (June 1, 2020, to December 31, 2021) across the participating centers, resulting in a cohort of 107 patients. We acknowledge that this sample size may limit the exploration of subgroup differences.
Point 3: did any of the patients have surgery or radiation therapy?
Response: We are grateful for the reviewer’s question about potential confounders. Our study design (Methods 4.1) proactively minimized their influence.
Specifically, we excluded patients who underwent surgery for the pain-causing lesion within the past week, or those starting or scheduled to start new molecularly targeted drugs or immune checkpoint inhibitors within 2 weeks before or 1 week after enrollment. While 31.8% of patients had ongoing anticancer treatment (including chemotherapy/radiation) at baseline (Table 1), crucially, no patients initiated any new anticancer therapies (including chemotherapy and radiation) during the 7-day observation period.
Furthermore, Results Section 2.5 demonstrates minimal impact from concomitant analgesics during this short period. Importantly, linear regression analysis revealed no significant correlation between changes in opioid doses (MEDD) and the reduction in worst pain intensity. Adjustments to other non-opioid analgesics and adjuvant analgesics (excluding corticosteroids) were also limited.
Given this limited influence from concomitant treatments, the rapid and substantial CR-NP improvement observed within just 7 days would suggest the initiated systemic corticosteroids were the primary driver of the analgesic effectiveness observed.
Point 4: why different corticosteroids were allowed?
Response: Thank you for the important question. The decision to allow different types of corticosteroids (dexamethasone, betamethasone, and prednisolone) was made to reflect real-world clinical practice, as this study aimed to observe the effectiveness of these agents as they are typically used by physicians based on their clinical judgment and patient factors. To standardize the analysis across different agents and doses, all corticosteroid doses were converted to a Dexamethasone Equivalent Daily Dose (DEDD), as described in the Methods section (4.3 and 4.4). We acknowledge that this heterogeneity in treatment is a limitation, as noted in our Discussion. We have added a sentence to the Methods section (page 12) as shown below, to clarify the rationale for allowing different corticosteroids.
Page 12:
To reflect real-world clinical practice, the protocol permitted the use of different systemic corticosteroids (dexamethasone, betamethasone, and prednisolone) based on physician preference and patient-specific factors.
Point 5: how many centers participated in the study?
Response: Thank you for your question. This study was conducted across 17 medical facilities in Japan. This information is stated in the Methods section (4.1. Study Design, Page 11) and also mentioned in the Discussion.

Reviewer 2 Report
Comments and Suggestions for Authors
This is a very interesting multicenter, prospective study assessing pain and daily activity impairment of inpatients with cancer-related neuropathic pain (CR-NP) who initiated or escalated corticosteroid therapy.
Comments:
Page 12, lines 398-399: “CR-NP was diagnosed based on imaging—computed tomography (CT), magnetic resonance imaging (MRI), or positron emission tomography-computed tomography (PET-CT)—and physical assessments by attending physicians.“ Could the authors please give more detailled transparency how exactly the diagnosis of “neuropathic pain“ was determined?
Table 2: What is somewhat unexpected that all patients were diagnosed with neuropathic pain, however, only 26% received gabapentinoids and only 5% antidepressants which appears unusual; pease explain.
Page 3, lines 117-118: “ Among the 107 patients at T0, CR-NP was categorized as follows: 27 patients with malignant brain tumors, 7 with leptomeningeal carcinomatosis, 22 with spinal cord tumors involvement, 30 with radiculopathy, and 21 with peripheral nerve involvement.“ Central and peripheral neuropathic pain are very distinct entities, thus, it would be very interesting whether a subgroup analysis would reveal some interesting differences.
It would be also important to know whether certain hospitals contributed most to the investigated patient population and whether this might have had an impact?
Since there was a potential dose escalation over the time period of investigation, it would be important to know whether (particularly from T1 until T2) and how much the corticosteroid dose were escalated and wether dose escalation itself was contributing to increased reduction in pain intensity? The same information is important for the opioid medication.
Another confounding factor is that pain intensity and impaired daily activity apparently were always assessed retrospectively, i.e. over the last 24 h, which should be addressed as a limitation in the discussion. In addition, with a lack of a patient group without steroid treatment, results could be due to the natural corse of disease; please address this under limitations.
Moreover, the authors should give the information (if available) about the highest daily blood sugar level.
How many patients per group would the authors envision from their data to perform a hypothesis driven clinical investiagtion in the future?
Author Response
We are grateful for the reviewer’s comments on our manuscript and believe that incorporating the advice therein into the revised version has improved the manuscript.
Our alterations to the manuscript are listed in the following point-by-point responses to the reviewers’ comments. In addition, we have rechecked our manuscript for conformance with the formatting guidelines (including References) and corrected some language issues (The reviewer’s comments are in blue font, and the revised text in the manuscript is in red font).
Point 1: Page 12, lines 398-399: “CR-NP was diagnosed based on imaging—computed tomography (CT), magnetic resonance imaging (MRI), or positron emission tomography-computed tomography (PET-CT)—and physical assessments by attending physicians.“ Could the authors please give more detailled transparency how exactly the diagnosis of “neuropathic pain“ was determined?
Response: Thank you for the important point regarding diagnostic transparency. The 'clinical assessment by attending physicians', conducted alongside imaging, guided the subsequent application of the specific diagnostic criteria. This also allowed physicians to incorporate pertinent physical findings—including neurological signs where neuropathic pain was clinically indicated—to contribute to the overall CR-NP diagnostic picture.
To improve clarity and accurately reflect this process, we have added the sentence (in red font) as below.
Page 12:
CR-NP was diagnosed based on imaging—computed tomography (CT), magnetic resonance imaging, or positron emission tomography-CT—and physical assessments by attending physicians, incorporating pertinent physical findings indicative of neuropathic pain, followed by application of etiology-specific diagnostic criteria [8].
Point 2: Table 2: What is somewhat unexpected that all patients were diagnosed with neuropathic pain, however, only 26% received gabapentinoids and only 5% antidepressants which appears unusual; pease explain.
Response: Thank you for your comment regarding the baseline adjuvant analgesics usage in Table 2. To clarify, this data reflects patient status at enrollment (T0), precisely. The information in Table 2 was not necessarily a point long after an established CR-NP diagnosis for all participants but also the information before corticosteroid treatment for CR-NP was started. Therefore, the baseline adjuvant analgesics usage simply reflects the varied medication status of patients at this specific timepoint for various reasons. We hope this clarifies the data.
Point 3: Page 3, lines 117-118: “ Among the 107 patients at T0, CR-NP was categorized as follows: 27 patients with malignant brain tumors, 7 with leptomeningeal carcinomatosis, 22 with spinal cord tumors involvement, 30 with radiculopathy, and 21 with peripheral nerve involvement.“ Central and peripheral neuropathic pain are very distinct entities, thus, it would be very interesting whether a subgroup analysis would reveal some interesting differences.
Response: Thank you for suggesting a subgroup analysis comparing central and peripheral CR-NP. We agree that these are distinct entities. This analysis has been already performed, categorizing patients based on the five specific oncologic etiologies of CR-NP (malignant brain tumors, leptomeningeal carcinomatosis, spinal cord involvement, radiculopathy, and peripheral nerve involvement). The results, showing changes in pain intensity for each subgroup, are presented in Figure 2 and described in the Results section (2.4. Changes in CR-NP Intensity by Pathological Mechanism). We also discussed these findings, noting that CR-NP linked to CNS or spinal cord involvement appeared to show more pronounced pain relief compared to CR-NP associated with peripheral nerve involvement.
Point 4: It would be also important to know whether certain hospitals contributed most to the investigated patient population and whether this might have had an impact?
Response: Thank you for raising the important point about potential center effects in our multicenter study. We agree that factors such as uneven patient distribution across the 17 participating sites or center-specific practices could potentially influence outcomes in observational research.
We have added a sentence acknowledging this to the Limitations section of our Discussion.
Page 11:
Additionally, we did not assess potential center-specific effects across the 17 participating centers on treatment patterns and outcomes, acknowledging this as a further limitation inherent to multicenter observational design.
Point 5: Since there was a potential dose escalation over the time period of investigation, it would be important to know whether (particularly from T1 until T2) and how much the corticosteroid dose were escalated and wether dose escalation itself was contributing to increased reduction in pain intensity? The same information is important for the opioid medication.
Response: Thank you for this insightful question. Our planned analysis for this prospective observational study focused primarily on evaluating the overall changes from baseline (T0), as well as describing the general trends in medication. A specific subgroup analysis isolating the effects of dose escalations or other adjustments during observational periods was not part of our original analysis plan for this study.
However, we certainly agree with your point that, even though the overall median doses tended to decrease or stabilize, we could not exclude the possibility that a subgroup of patients requiring dose escalation might have experienced more favorable pain relief. We have added a sentence acknowledging this point to the Limitations section of our Discussion as below.
Page 11:
Furthermore, although overall median doses of corticosteroids tended to decrease or stabilize between T1 and T2, we cannot exclude the possibility that a subgroup of patients requiring dose escalation might have experienced more favorable pain relief.
Point 6: Another confounding factor is that pain intensity and impaired daily activity apparently were always assessed retrospectively, i.e. over the last 24 h, which should be addressed as a limitation in the discussion. In addition, with a lack of a patient group without steroid treatment, results could be due to the natural corse of disease; please address this under limitations.
Response: Thank you for highlighting these important methodological considerations as potential limitations.
We certainly agree this is a critical consideration when interpreting results from studies lacking a control group. This inherent challenge – distinguishing the specific treatment effect from changes potentially due to the natural course of the disease or placebo effects – is closely related to the primary limitation discussed at the beginning of our Limitations section.
In addition, we agree that relying on patient recall over the preceding 24 hours for pain and interference ratings introduces potential for recall bias. We will explicitly acknowledge this by adding the following statement at the Limitations section as below:
Page 11-12:
Lastly, the reliance on patient recall over a 24-h period to report pain relief and PROs may have introduced recall bias, affecting the accuracy of the data.
Point 7: Moreover, the authors should give the information (if available) about the highest daily blood sugar level.?
Response: Thank you for inquiring about more detailed blood sugar data. As described in our Methods (section 4.3), adverse events, including hyperglycemia, were systematically monitored and graded according to the standardized CTCAE v5.0 criteria. Our study protocol was designed to capture adverse events based on this established grading system, rather than collecting specific numerical data points such as the highest daily blood sugar level for each participant. Therefore, unfortunately, this specific granular data is not available from our collected dataset as it was outside the pre-defined scope of data collection.
Point 8: How many patients per group would the authors envision from their data to perform a hypothesis driven clinical investiagtion in the future?
Response: Thank you for this forward-looking question regarding the design of future hypothesis-driven studies. Determining the precise sample size per group would indeed require specifying key parameters for that future trial, such as the primary endpoint, the clinically meaningful effect size being targeted, the desired statistical power, and the significance level. While calculating a specific sample size is beyond the scope of the current manuscript, we agree that the data presented herein, particularly the observed magnitude and variability of the changes in pain scores, provides essential foundational information. Our outcomes would be crucial for informing accurate power calculations when designing such future clinical investigations.

Reviewer 3 Report
Comments and Suggestions for Authors
This is a well-executed multicenter, prospective observational study that provides valuable data on the analgesic and functional effects of systemic corticosteroids in managing cancer-related neuropathic pain (CR-NP). The authors have successfully addressed a clinically important and under-investigated area, especially in the palliative care context.
I just have minor points
Duration of Observation: A 7-day observation period is relatively short. While this is acknowledged, it would be useful to provide more detail on the potential trajectory of corticosteroid effectiveness beyond this period.
Data Interpretation Caveats: The manuscript might further stress that the observed improvement in pain and ADL could also be partially attributable to placebo effects or natural symptom fluctuation in the absence of a control group.
Terminology Consistency: There is some redundancy and variability in terminology, such as “pain reduction” and “analgesic benefit.” Harmonizing terminology would improve readability.
Comments on the Quality of English LanguageThe manuscript is written in generally good English, but a few phrases could be restructured for improved clarity. A final round of professional proofreading (especially in the Methods and Discussion sections) would improve the manuscript.
Author Response
We are grateful for the reviewer’s comments on our manuscript and believe that incorporating the advice therein into the revised version has improved the manuscript.
Our alterations to the manuscript are listed in the following point-by-point responses to the reviewers’ comments. In addition, we have rechecked our manuscript for conformance with the formatting guidelines (including References) and corrected some language issues (The reviewer’s comments are in blue font, and the revised text in the manuscript is in red font).
Point 1: Duration of Observation: A 7-day observation period is relatively short. While this is acknowledged, it would be useful to provide more detail on the potential trajectory of corticosteroid effectiveness beyond this period.
Response: Thank you for your comment regarding the 7-day observation period and your interest in the longer-term effectiveness trajectory. We agree that understanding the duration of effect beyond one week is clinically very relevant. As we explicitly state in our Limitations section, this study was specifically designed to assess the rapid analgesic and functional benefits within the first 7 days. Therefore, providing detailed data or predictions about the trajectory beyond this period based solely on our current findings would unfortunately be speculative and exceed the scope of our reported outcomes.
Clinically, while challenges with cumulative adverse effects may limit effectiveness over many months, it seems that effectiveness would be maintained over several weeks, provided that careful dose adjustments are made to balance efficacy and tolerability. Our current 7-day study could not provide data to support this over 7-day outcomes, which remains an area for future exploration.
Point 2: Data Interpretation Caveats: The manuscript might further stress that the observed improvement in pain and ADL could also be partially attributable to placebo effects or natural symptom fluctuation in the absence of a control group.
Response: Thank you for emphasizing the need to consider potential placebo effects and natural symptom fluctuation – a crucial point also underscored by other reviewers, highlighting its importance in interpreting our findings from this uncontrolled study. This inherent challenge of distinguishing specific treatment effects from other factors is indeed the central reason behind the primary limitation stated at the very beginning of our Discussion's Limitations section. As articulated there, our observational design, lacking a placebo comparison group, prevents definitive conclusions about causality and the exclusion of influences like the natural disease course or placebo responses.
Therefore, while our upfront acknowledgment addresses this fundamental limitation, we recognize that definitively quantifying the true treatment effect separate from these factors would necessitate a RCT design. We believe our current observational findings provide crucial preliminary data that can inform the rationale and design for such future, more definitive studies.
Response to Reviewer 3, Point 3 & Comments on the Quality of English Language:
Point 3: Terminology Consistency: There is some redundancy and variability in terminology, such as “pain reduction” and “analgesic benefit.” Harmonizing terminology would improve readability.
Comments on the Quality of English Language: The manuscript is written in generally good English, but a few phrases could be restructured for improved clarity. A final round of professional proofreading (especially in the Methods and Discussion sections) would improve the manuscript.
Response: We appreciate your feedback on the clarity of our manuscript in terms of language. Despite an initial proofreading by a professional English language editing service (Editage; https://www.editage.com/services/english-editing), we acknowledge the necessity of further refinement. Accordingly, we have arranged for an additional round of language editing to enhance the readability and clarity of the text with the English language editing service (the revised text in the manuscript is in red font).
In addition, the references are re-formatted based on the journal guidelines, Cancers, but we have not marked these edits and revises in red font (because those are mostly related to case of the words).

Round 2
Reviewer 1 Report
Comments and Suggestions for Authors
Thanks for the revised ms.
Author Response
We are grateful for the opportunity to revise our manuscript and believe that incorporating the constructive feedback which has significantly improved our manuscript.
Our alterations to the manuscript are listed in the following point-by-point responses to the reviewers’ comments. In addition, we have rechecked our manuscript for conformance with the formatting guidelines (including References) and corrected some language issues (The reviewer’s comments are in blue font).
Reviewer: 1
Thanks for the revised ms.
Response: We deeply appreciate your reviewing our revised manuscript and your positive feedback.

Reviewer 2 Report
Comments and Suggestions for Authors
The manuscript has been improved according to the authors changes in the manuscript, however, the statements in the abstract and conclusions are still too strong in light of the fact that there was no control or placebo group which should be clearly mentioned in both abstract and conclusions.
Author Response
We are grateful for the opportunity to revise our manuscript and believe that incorporating the constructive feedback which has significantly improved our manuscript.
Our alterations to the manuscript are listed in the following point-by-point responses to the reviewers’ comments. In addition, we have rechecked our manuscript for conformance with the formatting guidelines (including References) and corrected some language issues (The reviewer’s comments are in blue font, and the revised text in the manuscript is in red font).
Reviewer: 2
The manuscript has been improved according to the authors changes in the manuscript, however, the statements in the abstract and conclusions are still too strong in light of the fact that there was no control or placebo group which should be clearly mentioned in both abstract and conclusions.
Response: We sincerely thank the reviewer for their careful re-evaluation of our revised manuscript.
We agree with the reviewer regarding the importance of contextualizing the findings presented in the Abstract and Conclusions given the study's observational design. We acknowledge that this context is crucial for the appropriate interpretation of the study's outcomes.
Specifically, we have now explicitly mentioned the study type and qualified the concluding statement in Abstract and Conclusion (in red font) as follows.
Page 2 (Abstract):
This multicenter, prospective observational study enrolled in patients with CR-NP who initiated or escalated corticosteroid therapy.
Page 2 (Abstract):
Corticosteroids provided rapid and considerable analgesic and functional benefits for patients with CR-NP in this observational setting; further validation through comparative controlled studies is required.
Page 15 (Conclusions)
Further studies, particularly randomized controlled RCTs incorporating control or placebo groups, are necessary to validate these findings, define optimal dosing, assess improvements in QOL, and ensure both effectiveness and safety.
